# Integrative RNA-Seq and H3 Trimethylation ChIP-Seq Analysis of Human Lung Cancer Cells Isolated by Laser-Microdissection

**DOI:** 10.3390/cancers13071719

**Published:** 2021-04-05

**Authors:** Quang Ong, Shingo Sakashita, Emi Hanawa, Naomi Kaneko, Masayuki Noguchi, Masafumi Muratani

**Affiliations:** 1Doctoral Program in Biomedical Sciences, Graduate School of Comprehensive Human Sciences, University of Tsukuba, Tsukuba, Ibaraki 305-8575, Japan; s1736043@s.tsukuba.ac.jp; 2Department of Genome Biology, Faculty of Medicine, University of Tsukuba, Tsukuba, Ibaraki 305-8575, Japan; emi.hanawa@tecan.com (E.H.); bisui25nao@gmail.com (N.K.); 3Department of Diagnostic Pathology, Faculty of Medicine, University of Tsukuba, Tsukuba, Ibaraki 305-8575, Japan; ssakashi@east.ncc.go.jp; 4Transborder Medical Research Center, University of Tsukuba, Tsukuba, Ibaraki 305-8575, Japan

**Keywords:** RNA-Seq, ChIP-Seq, NSCLC, H3K4me3, laser-microdissection, lung cancer, integrative analysis, tissue heterogeneity, transcriptomic, epigenomic

## Abstract

**Simple Summary:**

Tissue heterogeneity is one of the major problems in cancer genomics. Thus, we developed and conducted an RNA-Seq and ChIP-Seq integrative analysis of clinical lung tissue samples with the isolation of specific cell populations using laser-microdissection microscopy (LMD). The transcriptomic profile was successfully captured and somatically altered regions marked by histone H3 lysine 4 trimethylation (H3K4me3) were identified in lung cancer. We also observed the differential expressions of cancer-related genes near the altered proximal H3K4me3 regions, while altered distal H3K4me3 regions were overlapped with enhancer activity annotations of cancer regulatory genes. Additionally, proximal tumor-gained promoters were associated with the core components of polycomb repressive complex 2. Our study demonstrates the practical workflow of using LMD on clinical samples for integrative analyses, which improves the overall understanding of genetic and epigenetic dysregulation of malignancy.

**Abstract:**

Our previous integrative study in gastric cancer discovered cryptic promoter activation events that drive the expression of important developmental genes. However, it was unclear if such cancer-associated epigenetic changes occurred in cancer cells or other cell types in bulk tissue samples. An integrative analysis consisting of RNA-Seq and H3K4me3 ChIP-Seq was used. This workflow was applied to a set of matched normal lung tissues and non-small cell lung cancer (NSCLC) tissues, for which the stroma and tumor cell parts could be isolated by laser-microdissection microscopy (LMD). RNA-Seq analysis showed subtype-specific differential expressed genes and enriched pathways in NSCLC. ChIP-Seq analysis results suggested that the proximal altered H3K4me3 regions were located at differentially expressed genes involved in cancer-related pathways, while altered distal H3K4me3 regions were annotated with enhancer activity of cancer regulatory genes. Interestingly, integration with ENCODE data revealed that proximal tumor-gained promoters were associated with EZH2 and SUZ12 occupancies, which are the core components of polycomb repressive complex 2 (PRC2). This study used LMD on clinical samples for an integrative analysis to overcome the tissue heterogeneity problem in cancer research. The results also contribute to the overall understanding of genetic and epigenetic dysregulation of lung malignancy.

## 1. Introduction

Gene expression abnormality is one of the main characteristics of cancer cells and is regulated at multiple levels. At the genomic level, many studies have taken great efforts to understand changes in cancer gene expressions by identifying the genetic mutations. Some of the well-characterized genes carrying mutations include *TP53*, *RB1*, *EGFR*, and *KRAS*, which are frequently mutated in various cancer types, whereas others are rare and/or restricted to one cancer [1]. Distinct from genetic mutations, epigenetic regulations refer to modifying gene expression without permanent changes in the genomic sequence. They are preferentially applied in cancer cells given that epigenetic alterations are reversible and regulated faster compared to genomic evolution [2]. Advances in high-throughput technologies have enabled integrative genomic and epigenomic analyses, which have unraveled many novel molecular aberrations and network alterations in cancer. In particular, our previous integrative study in gastric cancer discovered cryptic promoter activation events that drive the expression of important developmental genes [3]. However, it was unclear if such cancer-associated epigenetic changes occurred in cancer cells or other cell types in bulk tissue samples. The use of bulk tissue, which is not precisely characterized in terms of histology, has long been the basis for molecular analysis. Undoubtedly, this approach is insufficient for a detailed analysis of molecular alterations, which might be restricted to a specific cell population such as tumor, normal, stroma, or epithelial cells.

In this study, we developed a workflow for an integrative RNA-sequencing (RNA-Seq) and chromatin immunoprecipitation sequencing (ChIP-Seq) analysis of clinical tissue samples in combination with the isolation of specific cell populations by laser-microdissection microscopy (LMD). A contamination-free and very exact selection is the main advantage of LMD and the published use of microdissection has enormously increased in the last decade [4]. As a pilot study, we applied this workflow to a set of lung adenocarcinoma (LUAD) and lung squamous cell carcinoma (LUSC) cases in which matched normal lung tissue, stroma cell, and tumor cell parts can be isolated. LUAD and LUSC are the most common subtypes of non-small cell lung cancer (NSCLC) [5]. Despite the recent advances in NSCLC therapies, the high mortality of NSCLC patients has not significantly decreased over the years [6]. Thus, it is important to precisely elucidate the mechanisms involved in NSCLC at molecular levels to develop more effective and safe treatment strategies. However, most lung tumors have supporting stroma cells including infiltrating inflammatory cells, neovascular cells, and fibroblasts [7]. These adjacent nonmalignant components contribute to the lung tumor microenvironment, while certainly having a distinct gene expression profile from the tumor cell part [8].

We hypothesized that our study could provide more precise gene expression and histone modification profiles by enhanced cell selection using LMD for clinical NSCLC tissue samples. Therefore, first, we performed RNA-Seq for gene expression profiling. As expected, the transcriptomic profile of NSCLC was successfully captured, and confirmed the well-known significant differential expressed genes (DEGs) and enriched pathways of two subtypes of NSCLC, LUAD and LUSC. Moreover, comparison with the Cancer Genome Atlas (TCGA) (https://tcgadata.nci.nih.gov/tcga, accessed on 20 July 2019) suggested that our RNA-Seq results were not only representative for NSCLC, but also more precise by enhanced cell selection using LMD. Next, we examined epigenetic alteration in NSCLC by using ChIP-Seq on histone H3 lysine 4 trimethylation (H3K4me3) marks. Altered H3K4me3 may represent abnormal epigenetic control that can cause changes in cancer-associated gene expression and in the regulation of fundamental cancer-associated functions including growth and metastasis [9,10]. By comparing the H3K4me3 profile of lung tumor and normal tissues, we identified hundreds of somatically altered promoters. Interestingly, we observed cancer-related DEGs located at proximal H3K4me3 regions, while altered H3K4me3-marked promoters distant from transcription start sites (TSSs) were annotated with enhancer activity of cancer regulatory genes. Moreover, integration with ENCODE data revealed that proximal tumor-gained promoters were associated with EZH2 and SUZ12 occupancies, which are the core components of PRC2 (polycomb repressive complex 2).

Taken together, this study demonstrates a practical workflow that uses LMD on clinical samples for an integrative RNA-Seq and ChIP-Seq analysis, which highlights the gene and pathway signatures as well as the annotation of somatically altered H3K4me3 regions in NSCLC. Our results contribute to the overall understanding of genetic and epigenetic dysregulation of lung malignancy.

## 2. Materials and Methods

### 2.1. Tissue Samples and Laser-Microdissection Microscopy (LMD) Isolation

Post-surgical tissue specimens stored in the Pathology Department and Tsukuba Human Tissue Biobank at the University of Tsukuba Hospital (Ibaraki, Japan) were used. A normal tissue portion was collected at the same time as the lung cancer tissue portion from each post-surgical material. The lung cancers examined were classified according to the World Health Organization (WHO) classification (4th ed.) [11]. Age and gender information of patients from whom the samples were collected are provided in Appendix A with the range and mean values for age.

### 2.2. RNA-Sequencing

Fresh-frozen tissue samples were partially thawed in 1% formaldehyde/PBS (Phosphate-buffered saline) and continued to fix for 10 min at room temperature. Fixed tissue was rinsed in PBS and embedded in OCT (optimal cutting temperature) compound (4583 Tissue-Tek, Sakura Finetek, Tokyo, Japan). Tissue samples were sectioned (10 μm) and placed on membrane slides (Pen-membrane, Leica, Tokyo Japan). Sections were stained with hematoxylin and dehydrated by ethanol. Tissue parts were obtained by LMD (LMD6000, Leica) [12]. For the isolation of total RNA, sections were processed with an RNAeasy FFPE Kit (73504, Qiagen, Hilden, Germany). RNA yield and degradation status were monitored by a Bioanalyzer RNA Pico Kit (5067–1513, Agilent Technologies, Santa Clara, CA, USA). Total RNA samples (quantities in ng; Normal: range = 10.5–64.7, mean = 33.6; Tumor: range = 2.2–98.2, mean = 42.7; Stroma: range = 4.2–110, mean = 28.5) were converted into sequencing libraries using a SMARTer Stranded Total RNA Sample Prep Kit (635005, Takara Bio, Shiga, Japan). Sequencing was performed with Illumina NextSeq500 (Illumina, San Diego, CA, USA) to obtain 2 × 36-base reads.

### 2.3. RNA-Seq Analysis

As a pilot study, we first applied the LMD–RNA-Seq integrative approach to ten cases of NSCLC including six LUAD and four LUSC. For each case, normal lung tissue, stroma cell, and tumor cell parts were isolated by LMD. Typical scout slide images including representative margins outlined for microdissection are shown in Figure 1a. After sequencing, all RNA-Seq raw fastq data were mapped to the human reference genome (hg19, Genome Reference Consortium GRCh37) using Burrows–Wheeler Aligner (BWA) software [13] (version 0.7.12) and the ‘mem’ algorithm with the default settings (BWA-MEM). Annotation data were downloaded from the UCSC website (http://genome.uscs.edu, accessed on 8 June 2019). Raw counts were generated using featureCounts [14] (version 2.0.0) with ‘stranded’ and ‘paired-end’ reads options. Genes with less than 10 raw counts in every sample were eliminated from further analysis. Normalized counts were created, and significant DEGs between the tumor and normal samples were identified using the ‘DESeq2’ package [15]. An absolute fold change ≥ 2 and false discovery rate (FDR) adjusted *p*-value ≤ 0.01 were used to classify the DEGs. Heatmaps were generated using the ‘ComplexHeatmap’ package [16]. Pathway analysis was performed and plots were created using the ‘clusterProfiler’ package [17]. All packages were run on RStudio (https://rstudio.com/, accessed on 30 September 2018) with R version 3.6.3 (The R Project for Statistical Computing, Vienna, Austria).

Due to the small sample size, we intended to compare our RNA-Seq results with a bigger dataset from TCGA. Hence, LUAD and LUSC samples were downloaded from TCGA Data Portal (https://gdc.cancer.gov/, accessed on 20 July 2019) in the form of raw count files (file type: htseq.count.gz). At the time of conducting this analysis, there were a total of 1104 files available and retrieved from TCGA (512 files for LUAD and 512 files for LUSC). These files included ‘primary tumor’ and ‘solid tissue normal’ metadata descriptions from 980 lung cancer cases, which means that there were some cases with multiple tumor files as well as some cases without matched normal files. We then collected one representative tumor file and one matched normal file from those cases that had matched normal tissue files. Eventually, we had 101 TCGA cases (*n* = 52 for LUAD pairs and *n* = 49 for LUSC pairs) for further analysis (Appendix A). Normalization and DEGs analysis of TCGA data were the same as our RNA-Seq analysis described above.

General workflow of RNA-Seq analysis is shown in Appendix A.

### 2.4. Chromatin Immunoprecipitation Sequencing (ChIP-Seq)

LMD tissue parts from serial tissue sections were used for the Nano-ChIP assay after RNA purification [18]. Dissociated tissues were lysed in 200 μL lysis buffers (50 mM Tris-HCl pH8.0, 10 mM EDTA, 1% sodium dodecyl sulfate (SDS)) and sonicated (total 8 min, 30 s pulses with 30 s interval) using a Bioruptor (Diagenode, Toyama, Japan). Sonicated samples were diluted with 10 × volume of dilution buffer (10 mM Tris-HCl pH8.0), 140 mM NaCl, 1 mM EDTA, 1% Triton X-100, 0.01% SDS [18]) containing 0.1% SDS and precleared with Dynal Magnetic beads (Invitrogen, Life Technologies, Carlsbad, CA, USA). Chromatin solutions were centrifuged for 10 min at 10,000 r.p.m. and the supernatant was recovered. ChIP was performed overnight using K4me3 antibody (07–473, Millipore, 1 μL per ChIP) at 0.1% SDS and 140 mM NaCl. ChIP beads were washed with dilution buffer, then once with Tris-EDTA (TE) buffer and collected on a magnetic stand. Reverse crosslinking was performed with Pronase (Roche, Basel, Switzerland) at 42 °C for 2 h, then 68 °C for 6 h. After recovery of ChIP and input DNA by phenol–chloroform–isoamyl alcohol extraction and ethanol precipitation, DNA was used for sequencing library construction with the NEBNext Ultra II DNA Library Prep Kit for Illumina (E7645, New England Biolabs, Ipswich, MA, USA). Sequencing was performed with Illumina NextSeq500 (Illumina, San Diego, CA, USA) to obtain 2 × 36-base paired-end reads.

### 2.5. ChIP-Seq Read Mapping and Peaks Calling

For ChIP-Seq, we used tumor samples from six cases out of ten cases that were used in RNA-Seq. These six cases included three cases from squamous cell carcinoma (sample IDs: 13, 14, and 15) and three cases from adenocarcinoma (sample IDs: 16, 18, and 19). Due to the low number of cells in the normal samples collected from LMD (as shown in Appendix A), we could not obtain comparable ChIP-Seq data quality. Therefore, we generated a unified set of H3K4me3 ChIP-Seq signals for this study by merging six normal lung tissue H3K4me3 ChIP-Seq samples collected from the ENCODE database (https://www.encodeproject.org, accessed on 20 July 2019) with our six NSCLC samples. Information on these six cases is shown in Appendix A. ChIP-sequencing tags were mapped against the human reference genome (hg19) using Burrows–Wheeler Aligner (bwa) software [12] (version 0.7.12) and the ‘mem’ algorithm with the default settings. Uniquely mapped tags were used for peak calling by CCAT [19] version 3.0. Peak regions were filtered by the minimum number of read counts as 30 and the minimum score as 5. Peak regions from all tissue samples were pooled, and overlapping peak regions were merged to create a total set of peak regions for downstream analysis (representative views and descriptions of merging peaks are shown in Appendix A). We used the ENCODE blacklist [20] to remove a comprehensive set of problematic regions from the human genome. To quantify peak heights, we analyzed the ChIP-Seq data using featureCounts [13] (version 2.0.0). Raw read count values were estimated for our set of H3K4me3-marked promoter regions. Batch effects were adjusted, and significant differential ChIP-Seq signals were identified using the ‘DESeq2’ package [14] after TPM (Transcripts Per Kilobase Million) transformation of raw read counts. Heatmaps and plots were generated using the ‘ComplexHeatmap’ and ‘clusterProfiler’ packages [15,16], respectively. All packages were run on RStudio (rstudio.com) with R version 3.6.3. CAGE data of adult human lung tissue samples (https://fantom.gsc.riken.jp/5/sstar/FF:10019-101D1) were downloaded from the FANTOM5 database [21] for comparison. General workflow of ChIP-Seq analysis is shown in Appendix A.

### 2.6. Transcription Factor Binding Sites (TFBSs) Analysis (ENCODE)

Transcription factor (TF) binding regions were extracted from the transcription factor binding site (TFBS) cluster file (wgEncodeRegTfbsClusteredV3) downloaded from the UCSC genome browser [22], which corresponded to peaks called from the ChIP-Seq data for 338 TFs in 130 cell types performed by the ENCODE consortium [23]. We then intersected the H3K4me3 regions with the TF binding regions and determined the number of TFs found in each category of H3K4me3 sets (gained.TSS, loss.TSS, and altered.noTSS). Each overlapping TF was counted only once, regardless of the number of different cell types in which it was found. We then compared the number of occurrences of TFs between each of the above categories of the H3K4me3 and all H3K4me3 regions by Fisher’s exact test. The detailed workflow of this analysis is shown in Appendix A.

## 3. Results

### 3.1. Transcriptomic Profiles of Non-Small Cell Lung Cancer (NSCLC) Were Successfully Captured Using LMD-Isolated Samples

In total, more than 1 billion paired-end raw reads (36-bp in length) were generated for 30 samples, ranging from 32.9 to 53.2 million reads per sample (Appendix A). To align the clean reads to the reference genome, we employed the state-of-the-art and widely used aligner, BWA-MEM (see Section 2.3). On average, 91% of all the clean reads were aligned to the reference genome (Appendix A). As shown in Appendix A, the broad view of our RNA-Seq data on the UCSC browser suggested that we successfully obtained the gene expression signals for not only the tumor and normal samples, but also stroma samples. This improved our confidence to identify additional tissue-specific signals. For example, we observed that the *HOOK2* gene in case 19 was highly expressed in the tumor sample (19T), but not in its matched normal (19N) and stroma (19S) samples, while those genes around *HOOK2* (*SNORD41*, *TRIR*, and *JUNB*) had similar levels of expression in all three samples (Figure 1b). A summary of the RNA-sequencing and mapping of the 30 samples is presented in Appendix A. Principle component analysis (PCA) was performed with the normalized counts (based on the DESeq2 method) to investigate if samples from the same tissue type clustered together. We found normalized data formed three distinct clusters in the PCA-plot: normal, stroma, and tumor. As expected, tumor samples were correctly divided into two groups as these two lung cancer subtypes are reported to have distinct genomic profiles (Figure 1c). A total of 28,723 unique genes were detected after removing genes with raw count ≤10 in each tissue sample from the analysis. Figure 2a shows the heatmap that represents the 4118 DEGs between tumor versus normal tissues (see also Appendix A). For this analysis, we used both LUAD and LUSC in the tumor tissue group versus the normal group. However, heatmap clusters still presented the separation of gene expression patterns between these two NSCLC subtypes. On the other hand, the expression of stroma samples on the heatmap showed a sample-wise inconsistency, suggesting that these DEGs are significant for LMD-isolated tumor parts.

### 3.2. RNA-Seq Analysis Showed Subtype-Specific Differential Expressed Genes (DEGs) and Enriched Pathways in NSCLC

High quality RNA-Seq from LMD showed clear separation between cell types and NSCLC subtypes (Figure 2a). Therefore, we wanted to further investigate the differential gene expression profiles of LUAD (*n* = 6) and LUSC (*n* = 4). After applying a stringent filtering approach (see Section 2.3) that compared NSCLC subtypes with their matched normal samples (T vs. N), we identified 3211 DEGs of LUAD and 2771 DEGs of LUSC. Among the top highly expressed genes for each subtype (Figure 2b,c), *claudin 2* (*CLDN2*), *secretoglobin 3A2* (*SCGB3A2*), and *mucin 21* (*MUC21*) have been well reported in the literature as potential diagnostic biomarkers in lung adenocarcinoma [24,25,26]; meanwhile, keratin family members (*KRT6A*, *KRT6B*, *KRT6C*, *KRT14*, *KRT16*) have also been reported to be upregulated only in lung squamous cell carcinoma [27,28,29,30]. The details of the top 15 most up/downregulated genes of the two NSCLC subtypes ranked by fold change (FC) are provided in Appendix A. The KEGG pathway analysis on significant up-/downregulated genes revealed the differences and similarities between LUAD and LUSC (Figure 2d). Both NSCLC subtypes enriched for the biosynthesis of amino acids and carbon metabolism in upregulated pathways and focal adhesion and cell adhesion molecules (CAMs) in downregulated pathways. Other highly significant enriched pathways included mucin type O-glycan biosynthesis (upregulated in LUAD), cell cycle (upregulated in LUSC), cGMP-PKG signaling pathway (downregulated in LUAD), and viral myocarditis (downregulated in LUSC). The full results of the KEGG enriched pathway analysis including gene sets are listed in Appendix A. These results indicate that our RNA-Seq from 10 cases of LMD-isolated samples were good representatives for capturing gene expression profiles of NSCLC samples.

### 3.3. RNA-Seq Results from LMD-Isolated Tumor Samples Were Concordant with the Cancer Genome Atlas (TCGA) Reference Dataset While Excluding Stroma Parts

We then compared our RNA-Seq findings with those derived from the TCGA (https://gdc.cancer.gov/) for LUAD and LUSC versus matched normal sample comparisons. These RNA-Seq data were queried for tumor versus normal discrimination of individual transcripts and yielded 9748 and 13,402 significant genes (FDR-adjusted *p*-value ≤ 0.01, |fold change| ≥ 2) that were tumor discriminant in LUAD and LUSC, respectively. Almost all (58/60) of the top 15 up/downregulated genes of NSCLC subtypes in the present study (excluding pseudogenes) also exhibited nonzero readings in the TCGA RNA-Seq batch 101 dataset. The concordance in the qualitative direction of fold-change (up- or downregulated) between the present data (both LUAD and LUSC) and data from TCGA is further highlighted by the finding that 53/58 (91.4%) of overlapping genes were altered in the same direction (Appendix A). The remaining five genes were found to have statistically insignificant changes in the TCGA data. However, it is worth mentioning that the batch 101 TCGA data are all macroscopic (bulk) tissue samples, which include supporting stroma cells. Therefore, we decided to check the mRNA expression profile of our LMD-isolated stroma samples. We hypothesized that our LMD-isolated tumor samples were less contaminated as they were well separated from the stroma parts.

Using similar transcript count quantification and normalization methods, we found highly expressed genes in our stroma samples, some of which are typically used as marker genes for stroma cell subtyping. For example, *collagen type I alpha 1* (*COL1A1*) and *collagen type I alpha 2* (*COL1A2*) genes are commonly used as fibroblast and myofibroblast markers, while *immunoglobulin lambda constant 3* (*IGLC3*) and *immunoglobulin heavy constant gamma 3* (*IGHG3*) are marker genes for B cells. Noticeably, these genes showed minimal or low expression in our LMD-isolated tumor and normal samples, but their expressions in TCGA tumor samples were relatively high (Figure 2e,f). These results suggest that our NSCLC RNA-Seq data was not only representative of transcriptomic profiles, but also more precise by enhanced cell selection using LMD.

### 3.4. Somatically Altered H3K4me3-Marked Promoters in NSCLC Were Successfully Identified Using LDM-Isolated Samples

Even though there are established and distinct molecular subtypes of NSCLC as we described in the previous section, because of limited sample sizes, we elected in the current study to identify promoter alterations present in multiple NSCLC tissues relative to normal lung tissues irrespective of subtype. Focusing on recurrent alterations also had the benefit of capturing the relevant variations with relatively small sample size. The ChIP-sequencing produced an average of 56 million reads per sample for the six NSCLC cases of this study and 23 million reads per sample for the six ENCODE normal lung samples. Furthermore, over 93% of the reads were properly mapped to the human reference genome (range: 73–99%), which surpassed the ENCODE recommended guidelines for ChIP-Seq quality [31] (Appendix A). We successfully obtained genome-wide chromatin profiles for both normal and cancer tissues (Appendix A) and predicted 17,752 putative promoters marked by H3K4me3.

Comparison to FANTOM5 CAGE (Cap Analysis of Gene Expression)-defined promoters of lung tissues (see Methods) supported a large proportion of our H3K4me3-marked promoters (~88%). To identify somatically altered promoters in NSCLC, we quantified ChIP-Seq reads for the merged peak regions and compared normalized read counts between NSCLC and ENCODE normal lung samples. We identified 645 promoters exhibiting differential H3K4me3 modifications between NSCLCs and ENCODE normal lung tissues (tumor-associated promoters), of which 351 were tumor-gained promoters (upregulated H3K4me3 signal in tumor) (fold change ≥ 2 and FDR-adjusted *p*-value ≤ 0.05) and 294 were tumor-loss promoters (downregulated H3K4me3 signal in tumor) (fold change ≤ −2 and FDR-adjusted *p*-value ≤ 0.05) (Appendix A). Figure 3a is an example of a tumor-gained promoter (increased H3K4me3) at the *EPPK1* gene, a component of EGF (epidermal growth factor) signaling.

### 3.5. Proximal Tumor-Altered H3K4me3 Regions Were Found at Cancer-Related DEGs

We further analyzed the H3K4me3-marked promoters (hereafter referred to as “promoters”) with respect to the proximity to known genes and focused on the tumor-associated promoters. Then, we defined proximal H3K4me3 regions as those that overlapped with transcription start sites (TSSs) that mapped close (<1 kb) to TSS data retrieved from ENSEMBLE, one of the major transcript databases. About 70% (444/645) of tumor-associated H3K4me3-marked regions overlapped with TSS annotation (Figure 3b). Previous literature has reported that H3K4me3 mostly localized to the promoter-proximal region (1–2 kb from the TSS) and was permissive for active transcription [32,33]. We then analyzed the expression of genes that overlapped with tumor-associated promoters using RNA-Seq data previously shown in this study. Nearly 50% (313/645) of altered H3K4me3 regions were located at the 310 genes that were differentially expressed between tumor versus normal samples (Appendix A). Unsupervised clustering of the RNA-Seq data for these 310 genes showed an interesting observation between tumor-altered H3K4me3 regions and gene expression levels (Figure 3c). Particularly, most of the tumor-loss promoters (‘loss.TSS’), meaning loss of H3K4me3 signals in tumor tissues, showed higher gene expression in normal compared to tumor counterparts. In contrast, most genes with tumor-gained promoters (‘gained.TSS’) were highly expressed in tumor samples (especially, regarding their NSCLC subtypes). We next examined the pathways enriched by these DEGs associated with tumor-altered H3K4me3 histone marks. There were four cancer-related pathways significantly enriched by 24 genes (Figure 3d). These pathways consisted of the well-known PI3K-Akt signaling pathway and three other pathways associated with cell matrix adhesions in cancer: extracellular matrix (ECM) receptor interaction, cell adhesion molecules (CAMs), and focal adhesion (Appendix A). This pathway analysis result also highlights several DEGs associated with tumor-altered H3K4me3 histone marks including downregulated (tumor vs. normal) tumor suppressor genes: *Integrin alpha8* (*ITGA8*) [34], *Fms-like tyrosine kinase-1* (*FLT1*) [35], *junctional adhesion molecule 2* (JAM2) [36], *calcium voltage-gated channel auxiliary subunit alpha2delta2* (*CACNA2D2*) [37], and *dystrophin* (*DMD*) [38]; and upregulated potential oncogenes: *p-cadherin* (*CDH3*) [39], *claudin-1* (*CLDN1*) [40], *desmoglein-2* (*DSG2*) [41], *ephrin A3* (*EFNA3*) [42], *integrin subunit beta 4* (*ITGB4*) [43], *integrin subunit beta 8* (*ITGB8*) [44], and *laminin gamma 2* (*LAMC2*) [45].

### 3.6. Altered H3K4me3-Marked Promoters Distant from TSSs Were Annotated with Enhancer Activity of Cancer Regulatory Genes

Recent studies reported that intergenic H3K4me3 also marks enhancers [46]. We speculated that about 31% (201/645) of tumor-altered H3K4me3 regions, which do not overlap with TSSs (‘altered.noTSS’ promoters), may represent a subclass of enhancers in tumor samples. GeneHancer was recently introduced as a comprehensive database of human enhancers and their inferred target genes [47]. GeneHancer revealed that 152/201 (>75%) of ‘altered.noTSS’ promoters intersected with enhancers and then predicted the corresponding targeted genes (Appendix A). Comparison to our RNA-Seq data showed that 126 out of those targeted genes were differentially expressed (T vs. N). Gene ontology (GO) analysis for these DEGs suggested the most significant molecular function term was DNA-binding transcription activator activity (Appendix A). Noticeably, the gene list enriched for this GO term consisted of transcription factors, which were previously reported to be related to cancer progression and development. This includes oncogenes such as *high mobility group A2* (*HMGA2*) [48], *SRY-Box transcription factor 2* (*SOX2*) [49], *forkhead box A1* (*FOXA1*) [50]; and potential tumor suppressors such as *zinc finger protein 750* (*ZNF750*) [51], *GATA binding protein 2* (*GATA2*) [52], and *grainyhead like transcription factor 1* (*GRHL1*) [53]. We took a capture on the UCSC browser at the genomic region featuring the *HMGA2* gene as a representative example (Figure 3e). HMGA2 is a transcription factor that influences a variety of biological processes including the cell cycle process, DNA damage repair process, apoptosis, senescence, epithelial–mesenchymal transition, and telomere restoration. Overexpression of this gene has been observed in several cancers such as pancreas, gastric, lung, ovarian, breast, and other cancers [48]. In this study, *HMGA2* gene expression was significantly higher in the LUSC samples than matched normal samples, and tumor-gained H3K4me3 signals were found at the promoter and enhancer regions of this gene (Figure 3e). These results suggest that altered promoters distant from ENSEMBL TSS could be associated with the enhancer activity of cancer regulatory genes. However, further experiments are needed to test this hypothesis.

### 3.7. Proximal Tumor-Gained Promoters Associated with EZH2 and SUZ12 Occupancies and Enriched for Developmental Genes

To identify potential oncogenic mechanisms driving these tumor-associated promoters, we intersected the genomic locations of H3K4me3 marked promoters with transcription factor binding sites (TFBS) of 338 transcription factors from the ENCODE consortium [23]. Proximal tumor-gained promoters (‘gained.TSS’ regions) significantly enriched with EZH2 (*p*-value < 1.6 × 10–16, Fisher’s exact test) and SUZ12 (*p*-value < 3.5 × 10–15, Fisher’s exact test) binding sites (Figure 4a; Appendix A). Both EZH2 and SUZ12 are the core components of the polycomb repressor complex 2 (PRC2) epigenetic regulator complex [54]. This result is concordant with previous studies in gastric cancer, suggesting that cancer-associated promoters overlap with regions targeted by PRC2 epigenetic regulator complex and are particularly sensitive to EZH2 inhibition [3,55]. Many studies have shown that PRC2 can have both oncogenic and tumor-suppressive functions in a context-dependent manner [56]. Moreover, the core components, EZH2 and SUZ12, have been reported to have oncogenic roles independent of PRC2 in several cancer types including NSCLC [57,58].

In this study, *EZH2* gene expression was significantly upregulated in NSCLC samples (Figure 4b). In addition, we wanted to obtain more insight by examining those genes whose TSSs exhibited tumor-gained H3K4me3 signals and EZH2/SUZ12 TFBS. We then performed GO enrichment analysis for that gene list. Interestingly, the results showed the strong enrichment in many biological development processes (Figure 4c). Furthermore, the protein–protein interaction networks analysis of the gene list by STRING [59] suggests *EPCAM* is the primary functional interacting node that has the highest number of connections (Figure 4d). In this study, a H3K4me3-marked signal was found at the *EPCAM* promoter and positively correlated with high RNA-Seq expression in the tumor samples (Figure 4d). Additionally, EZH2 TFBS was also observed at *EPCAM* TSS (Figure 4e). Taken together, these results highlight the association between proximal tumor-gained promoters with EZH2 and SUZ12 occupancies, which potentially affects the expressions of developmental regulatory genes such as *EPCAM*.

## 4. Discussion

Our study demonstrated that we successfully applied LMD-isolated clinical samples for integrative RNA-Seq and ChIP-Seq analysis. First, in the RNA-Seq results, we were able to show the clear separation between the tumor, matched normal, and stroma cell parts with our RNA-Seq results (Figure 1b,c). Indeed, we successfully captured the most DEGs and their enriched pathways for NSCLC subtypes (Figure 2b–d). For instance, one of the most upregulated DEGs in LUAD of this study was *CLDN2*, which has been reported to be absent in normal lung epithelia, but highly expressed in human lung adenocarcinoma tissues and adenocarcinoma-derived cells including A549, RERF-LC-MS, and PC-3 cells [60]. Previous studies showed that the knockdown of *CLDN2* expression by small interfering RNA (siRNA) suppresses proliferation and invasion in A549 cells [24,60]. Likewise, the most upregulated genes in LUSC included a set of keratin family members (*KRT6A*, *KRT6B*, *KRT6C*, *KRT14*, *KRT16*). Several studies have provided evidence for active keratin involvement in cancer cell invasion and metastasis as well as in treatment responsiveness [61]. Additionally, *KRT6A* and *KRT6B* expression levels were previously reported to be higher in LUSC than in LUAD and were used as biomarkers for differentiating between these two subtypes [62]. Furthermore, our KEGG pathway analysis results showed some concordances with previous findings; for example, biosynthesis of amino acids and carbon metabolism have been proven to be important processes for cancer diagnostic and therapeutic approaches [63,64], while focal adhesion and cell adhesion molecules are crucial pathways for studying malignant transformation and metastasis, especially in lung cancer [65]. In a NSCLC subtypes manner, we found that the mucin type O-glycan biosynthesis pathway was only enriched for LUAD (and not for LUSC), consistent with a previous report of different expression patterns in this pathway between LUAD and LUSC [66]. Meanwhile, cell cycle related genes have been suggested as key upregulated genes in LUSC [67]. The identification of such pathways and genes simply emphasizes the principle that the precision of our RNA-Seq data comes from the precision of our cell-selection efforts using LMD.

As a pilot study with a limited number of samples, comparison with a bigger platform like TCGA gave us a better perspective on the correctness of our dataset. Our RNA-Seq results showed high agreement among the top DEGs with TCGA NSCLC samples, suggesting that our NSCLC gene expression profile was well-captured by LMD even with our small sample size. However, lung tumors have a decent amount of supporting stroma cells such as inflammatory cells, neovascular cells, and fibroblasts [8]. Since the LUAD and LUSC from TCGA are macroscopic (bulk) tissue samples, they could be expected to show a mixture of both subtypes in their DEGs, reflecting not only tumor but also stroma parts. Indeed, as examples, the high expressions of fibroblasts, myofibroblast markers (*COL1A1*, *COL1A2*), and B cell markers (*IGLC3*, *IGHG3*) were found in our LMD-isolated stroma samples and TCGA tumor samples, but not in our LMD-isolated tumor parts (Figure 2e,f). Noticeably, *COL1A1* and *COL1A2* are subtypes of Type I collagen, which is widely known to be produced by stroma fibroblast [68,69,70]. A number of previous studies have demonstrated that Type I collagen is important for cancer development [71,72,73,74]. Taken together, a major implication of these results is that published lung cancer-derived transcriptomes likely contain the confounding factor of cellular heterogeneity; in this case, the stroma cell part. Thus, our study demonstrates a possible solution for this issue by using the LMD isolation method.

The vast majority of human cancers harbor both genetic and epigenetic abnormalities, with fascinating interplay between the two [75]. Great effort has been devoted to understanding the role of histone modifications in the development of cancer. In particular, H3K4me3 is a histone modification with a well-documented association with gene activity. However, it remains an important challenge for locating and defining the biological activity of this regulatory element in cancer. Here, although we had a limited number of NSCLC samples profiled, we made several notable observations that improve our overall understanding of lung tumorigenesis. We found that out of our 645 TSS-proximal altered H3K4me3 regions, 50% (313) mapped to the 310 DEGs (Figure 3c, Appendix A). Previous studies have shown that 80% of the genes containing promoter-proximal H3K4me3 are actively transcribed in human stem cells [76], and genes with high levels of H3K4me3 tended to be highly expressed [77]. Interestingly, those DEGs associated with TSS-proximal altered H3K4me3 regions are enriched for cancer-related pathways. Our KEGG pathway enrichment analysis on the DEGs from our RNA-Seq results also included these pathways (Figure 2d, Appendix A) and we wanted to examine whether this result was significant or whether random subsets of DEGs would have given similar results. To answer this question, we conducted a simple test, in which we generated 1000 lists of 310 DEGs (same number of DEGs in Section 3.5) that were each randomly subsetted from the 4118 DEGs in Section 3.2 (Appendix A). We then ran the same KEGG pathway enrichment analysis for these 1000 gene lists as in Section 3.5. Appendix A shows the number of pathways that were outputted for each of the 1000 gene lists. We observed that 800 out of the 1000 gene lists (80%) gave no pathway (0). From there, the number of gene lists monotonically decreased as a function of increasing pathway count, beginning with 111 gene lists (11.1%) containing one pathway and ending with one gene list (0.1%) containing 10 pathways. Given that five pathways were discovered in Section 3.5, we can say that the chance of getting the same or larger number of pathways by chance would be less than 2% (20 out of 1000 randomized lists containing 310 DEGs). Appendix A shows the bar plot, in which we counted the percentage of occurrence of all 99 KEGG pathways that resulted in the enrichment analysis of 1000 gene lists. Each pathway had less than 5% chance (red dotted line) of occurrence in any given gene list. We also highlighted cancer-related pathways as blue bars. We zoomed in on the pathways with the highest chance of occurrence in any given gene list subset, and it contained two pathways that were similar to those in our Section 3.5 results. Even though these similar pathways were among the top-occurring ones in random subsets of gene lists, we can still confirm that the chance of random subsets of the DEGs in Section 3.2 being enriched in similar cancer-related pathways as seen in Section 3.5 was very low. Therefore, our finding of enriched cancer-related pathways for the list of DEGs in Section 3.5 was likely significant. Indeed, activation of the PI3K-Akt signaling pathway (Figure 3d) has been linked to enhanced stemness in cancer models [78]. Additionally, other pathways associated with the extracellular matrix (ECM) and adhesion in cancer were also significantly enriched (Figure 3d). ECM components have been perceived as important regulators in cancer progression. Abnormal ECM such as disrupted organization and changes in essential composition or topography of the ECM has been implicated in cancer initiation and metastasis by remodeling the behavior of stroma cells and promoting tumor-associated angiogenesis and inflammation [79]. Therefore, tumor-derived ECM components are important to help identify new prognostic or therapeutic targets. Indeed, a 29-gene ECM-related prognostic and predictive indicator has been introduced as a potential genomic tool to improve patient selection in early-stage NSCLC [80]. With the advantage of isolating tumor cells by LMD, these results from our study suggest a list of DEGs as potential tumor-derived ECM components in NSCLC (Appendix A).

Next, we noticed that only 31% (201/645) of tumor-associated promoters were outside of ENSEMBLE TSSs (‘altered.noTSS’ promoters). This observation appears to be different from our previous study in gastric cancer, which found a large proportion of cryptic cancer-associated promoters [3]. One possible reason for this dissimilarity could be because the genomic features were better annotated in lung tissue than in the gastric tissue samples. At the time of this writing, Gene Expression Omnibus (GEO) datasets (https://www.ncbi.nlm.nih.gov/gds) gave 78,780 results for the search of ‘human lung’ and only 22,335 for ‘human gastric’. For example, in this current study, which had better access to annotation, using GeneHancer [47], we found that 152/201 (>75%) of ‘altered.noTSS’ H3K4me3 regions intersected with enhancers and that these enhancers had 407 targeted genes (Appendix A). This finding is supported by recent studies, which reported that intergenic H3K4me3 also marks enhancers [81,82]. Additionally, 126 out of the 407 targeted genes of GeneHancer-predicted enhancers marked by our H3K4me3 were differentially expressed in tumor versus normal. These DEGs include several oncogenes and were significantly enriched in GO terms as transcription factors (Appendix A) such as *HMGA2* [48], *SOX2* [49], *FOXA1* [50], and tumor suppressors such as *ZNF750* [51], *GATA2* [52], and *GRHL1* [53]. These results are concordant with the finding of H3K4me3 super-enhancers in breast cancer tissues, which were found to be associated with transcription activity and cancer-related processes [46].

Furthermore, we observed the association between proximal tumor-gained promoters with EZH2 and SUZ12 occupancies (Figure 4a). This observation agrees with previous studies in gastric cancer [3,55], suggesting potential common roles of these PRC2 components in multiple cancer types. Although the epigenetic role of PRC2 is reported to be inhibited by the upregulated H3K4me3, the core components, EZH2 and SUZ12, have been shown to have oncogenic roles independent of PRC2 in several cancer types including NSCLC [57,58]. In our study, the expression of EZH2 is also upregulated in NSCLC samples (Figure 4b). Since EZH2 can regulate gene transcription [83,84], we decided to examine its potential target genes. Our Gene Ontology enrichment analysis suggests that the tumor-gained H3K4me3 regions exhibiting EZH2 and SUZ12 are enriched with genes related to development processes (Figure 4c). Among the top enriched genes, *EPCAM* was highlighted as the hub node in the protein–protein interaction analysis (Figure 4d). *EPCAM* was overexpressed in the majority of cancer tissues [85]. Aside from affecting intercellular adhesion, *EPCAM* influences other important functions relevant to tumor progression including cell proliferation and cancer stemness, which suggests an active role of *EPCAM* in cancer metastasis [86]. Additionally, the gene expression of *EPCAM* can be regulated by genetic, epigenetic, and transcriptional factors [87]. Even though the direct link between *EZH2* and *EPCAM* in cancer has not been reported, our results suggest a possible interaction of these two genes associated with somatic H3K4me3 signals in NSCLC (Figure 4e).

## 5. Conclusions

In summary, we integratively analyzed RNA-Seq and H3K4me3 ChIP-Seq data from clinical NSCLC samples isolated by LMD and showed the potential of this workflow through the perspective of human genome research. The sample size limited the power of our study. In the future, the current study could be improved with a larger cohort and by incorporating more types of omics data as well as different histone marks.

## Figures and Tables

**Figure 1 cancers-13-01719-f001:**
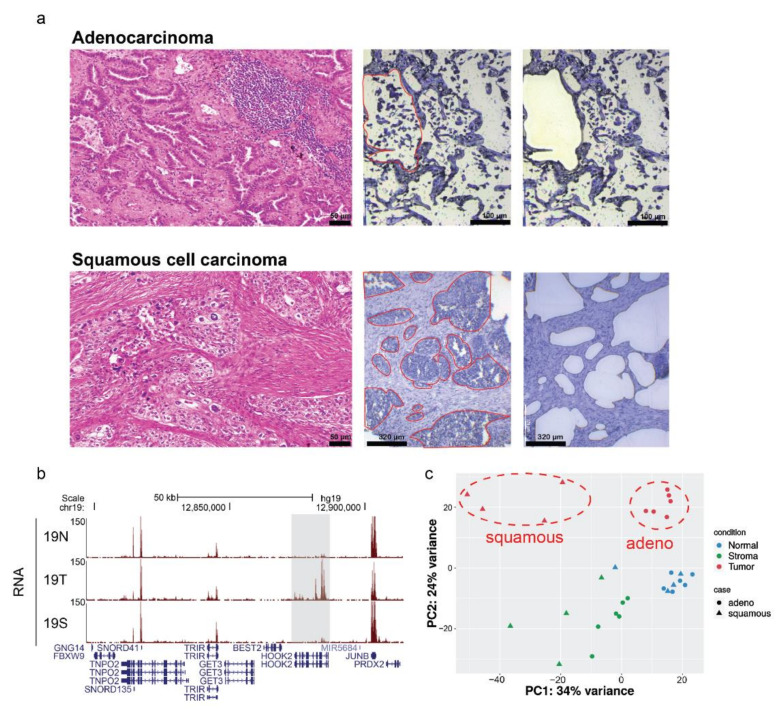
(**a**) Representative images of specimens undergoing laser microdissection (LMD). The examples of target areas are outlined with red line marks. The two lung cancer subtypes are adenocarcinoma (top row) and squamous cell carcinoma (bottom row). Scale Bar: 50 µm, 100 µm and 320 µm. (**b**) Representative view of RNA-Seq visualization captured using the UCSC genome browser. Case 19 is shown including the normal sample (19N), tumor sample (19T), and stroma sample (19S). The tumor-specific RNA-Seq signal is highlighted in the grey box. (**c**) PCA plot generated using the normalized counts (DESeq2 Methods). Lung tumor subtypes were separately clustered (red dotted circles).

**Figure 2 cancers-13-01719-f002:**
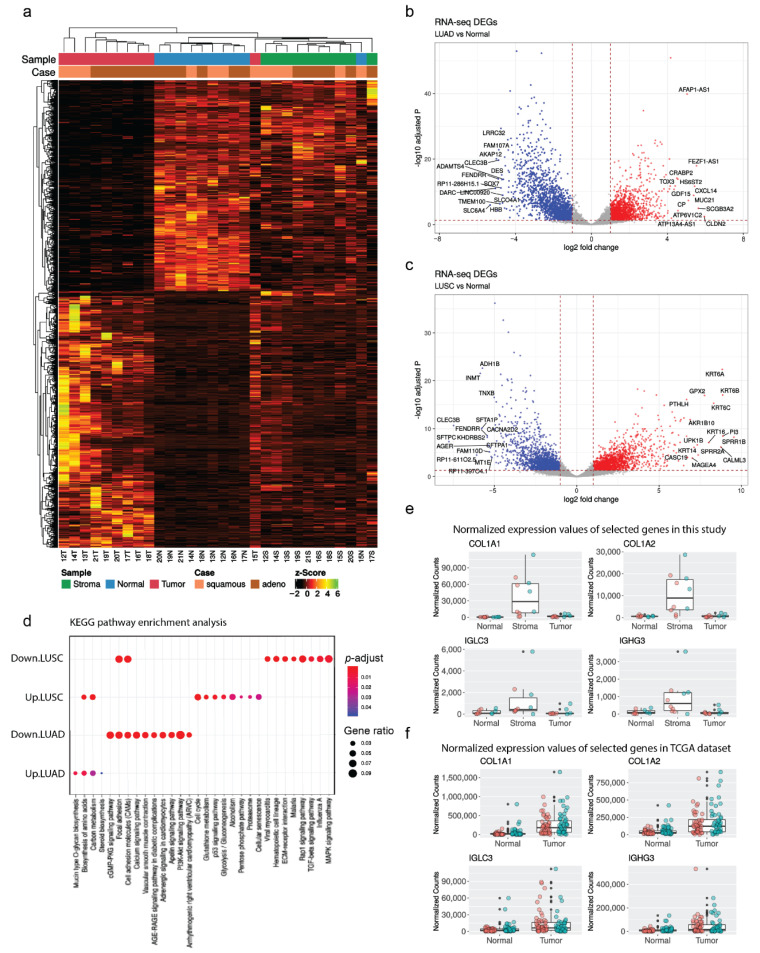
(**a**) Heatmap representation of 4118 genes of 30 samples (10 tumor, T; 10 normal, N; and 10 stroma, S) from 10 cases (Appendix A), filtered by FDR-adjusted *p*-value < 0.01 and absolute fold change ≥ 2. Volcano plot of DEGs between LUAD vs. normal (**b**), and LUSC vs. normal (**c**). With log2 (Fold change) as the x-axis and log10 (adjusted *p*-value) as the y-axis, the volcano plot was made according to the gene expression level. The red dots indicate upregulated genes, the blue dots indicate downregulated genes, and the grey dots indicate non-significant genes. The top 15 up/downregulated genes (ranked by fold change) highlighted with their gene names. (**d**) KEGG pathway enrichment analysis of significant up/downregulated genes in LUAD and LUSC. The color and size of the dots represent the range of adjusted *p*-value and the number of DEGs mapped to the indicated pathways, respectively. The top 10 enriched pathways are shown in the figure. Boxplot of mean expression levels for four indicated genes *COL1A1*, *COL1A2*, *IGLC3*, and *IGHG3* of: (**e**) this study’s normal samples (*n* = 10), stroma samples (*n* = 10), and tumor samples (*n* = 10); (**f**) TCGA data normal samples (*n* = 101), and tumor samples (*n* = 101). Individual values are presented as colored circles. Red circles indicate LUAD subtypes, light green circles indicate LUSC subtypes.

**Figure 3 cancers-13-01719-f003:**
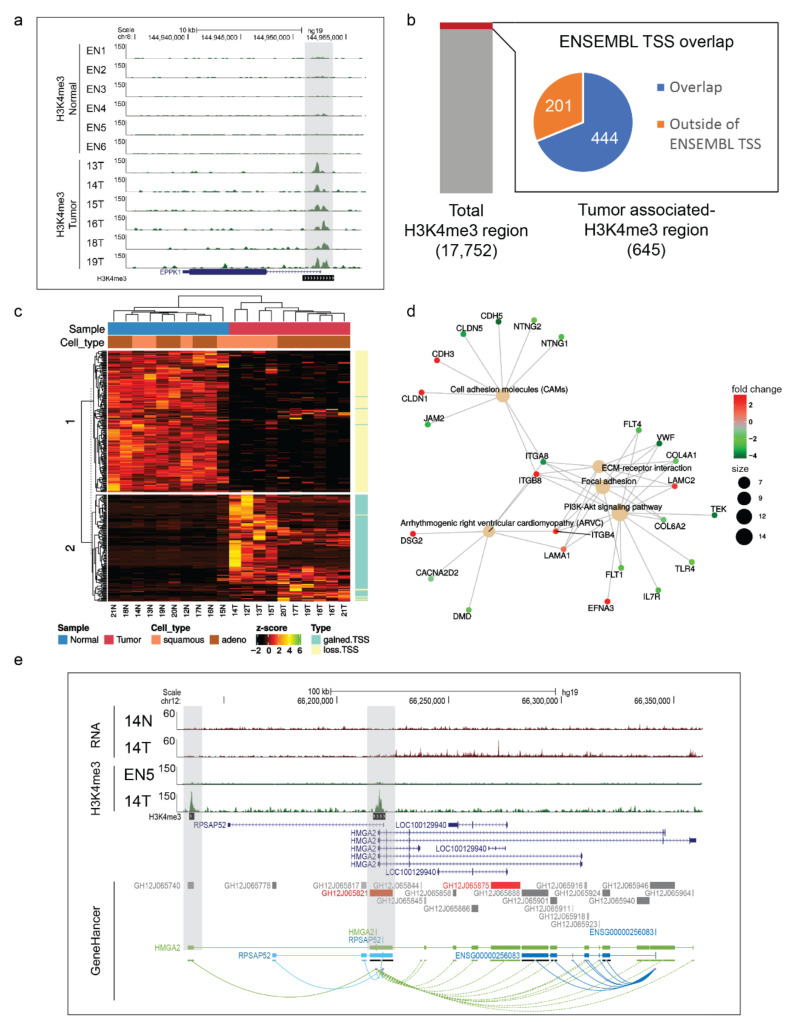
(**a**) Example of a tumor-gained function promoter. The UCSC genome tracks of the *EPPK1* TSS (grey shaded box) highlight the gain of H3K4me3 signals in NSCLC samples (13T, 14T, 15T, 16T, 18T, 19T) compared with ENCODE normal lung samples (EN1, EN2, EN3, EN4, EN5, EN6). (**b**) Determining overlaps between 645 tumor-altered H3K4me3 regions with ENSEMBL TSS (+/− 1 kb) regions. A total of 444 (69%) tumor-altered H3K4me3 regions overlapped while 201 (31%) were outside ENSEMBL TSS. (**c**) Heatmap of RNA-Seq read densities (row scaled) of 310 tumor vs. normal DEGs, whose TSSs intersected with tumor-altered H3K4me3 regions. (**d**) Cnet-plot from the KEGG pathway enrichment analysis of these 310 DEGs. KEGG pathways are shown with beige dots. (**e**) A capture of the UCSC genome illustrating the tumor-gained H3K4me3 signals (grey shaded boxes) at *HMGA2’s* TSS and potential enhancer regions defined by the GeneHancer database (GeneHancer IDs: GH12J065821 for the promoter, GH12J065740 for the enhancer). The two RNA tracks are RNA-Seq signals of tumor and normal samples of case 14. The two H3K4me3 tracks are ChIP-Seq signals of the case 14 tumor sample and ENCODE normal sample EN5.

**Figure 4 cancers-13-01719-f004:**
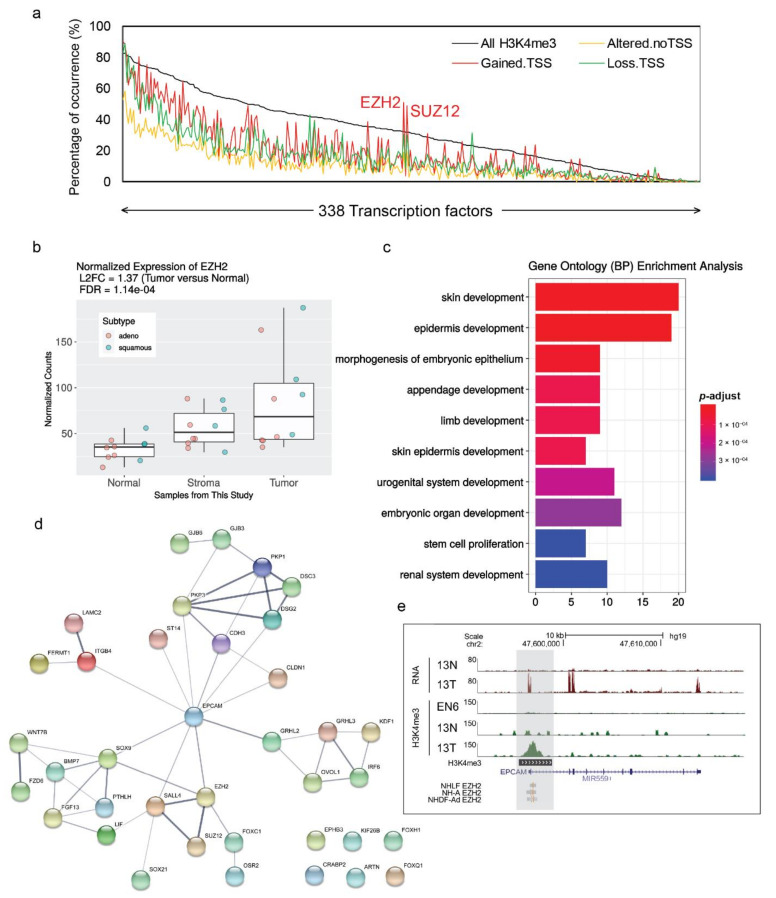
(**a**) Binding enrichment of ReMap-defined transcription factor binding sites at genomic regions exhibit H3K4me3 signals. Transcription factors were sorted according to their percentage of occurrences at all H3K4me3 regions. Proximal tumor-gained promoters (gained.TSS) significantly enriched with EZH2 and SUZ12 binding sites (*p*-value < 0.01, one-tailed Fisher’s exact test). The complete TF list is presented in Appendix A. (**b**) Boxplot of the median expression levels for *EZH2* in this study’s normal samples (*n* = 10), stroma samples (*n* = 10), and tumor samples (*n* = 10). Individual values are presented as colored circles. Red circles indicate LUAD subtypes, light green circles indicate LUSC subtypes. Log2-fold change (L2FC) and false-discovery rate (FDR) were calculated between tumor and normal samples. (**c**) Gene ontology (GO) terms for biological process (BP) enrichment analysis and (**d**) protein–protein interaction analysis from STRING of all the nearby genes of tumor-gained promoters exhibiting EZH2 and SUZ12 binding sites. (**e**) A capture of the UCSC genome illustrates the tumor-gained H3K4me3 signals (grey shaded boxes) at the TSS of the *EPCAM* gene, which exhibits the binding sites of EZH2 transcription factors. In this capture, samples from case 13 (normal, N; tumor, T) and ENCODE normal sample EN6 were used.

## Data Availability

Processed data and reproducible scripts for RNA-Seq and ChIP-SSeq analyses as well as all the figures generated by R are available online at https://github.com/QuangOngCM/LDM_RNAseq_ChIPseq_NSCLC.

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
