# Peer review of "Integrative RNA-Seq and H3 Trimethylation ChIP-Seq Analysis of Human Lung Cancer Cells Isolated by Laser-Microdissection"

_cancers, 2021, doi:10.3390/cancers13071719_

Round 1

Reviewer 1 Report

The authors conducted an RNA-seq and ChIP-seq integrative analysis to the specific cell types obtained from clinical lung tissues using laser-microdissection microscopy. The study is well designed, and the analysis pipeline looks good. Not only they found a correlation between changed gene with altered H3K4me3, but also, they compared their data with ENCODE data and found the involvement of EzH2 and Suz12.

Some minor comments:

For Materials and Methods

  1. If possible, it will be great if the authors provide more descriptions about the patients from whom tissue collected (for example, age, gender …) and how old were these tissues?
  2. How much RNA or DNA were used for library preparation?

For results,

It is good that the authors provide gene lists of DEG with changed H3K4me3, and top 15 DEG. It will be better to include gene lists for all the DEG and all changed H3K4me3, separately.

Also, all the raw data of RNA-seq and ChIP-seq should be deposit into NCBI GEO or somewhere else.

Figure 4b. Is there a statistic significant for these 2 gene changes?

Reviewer 2 Report

The authors present a combined RNA-seq and H3K4me3 ChIP-seq analysis using matched normal and non-small cell lung cancer tissues, for which stromal and tumor cells were isolated by laser-microdissection microscopy. They perform a standard gene expression analysis and highlight NSCLC subtype-specific markers. Based on the ChIP-seq/RNA-seq results, the authors find correlations between proximal (resp. distal) H3K4me3 regions and expression of nearby genes involved in cancer-related pathways (resp. enhancer activity of cancer-associated genes).

While there are interesting aspects, I am of the opinion that the paper requires more work, particularly from the methodological point of view and the presentation. The data and the results from this study should be made 
publicly available.

---------

1. The use of integrative RNA-seq and ChIP-seq analysis, and/or LMD is not new, see e.g.

https://pubmed.ncbi.nlm.nih.gov/30642670/
https://www.nature.com/articles/s41598-019-55146-2
https://plantmethods.biomedcentral.com/articles/10.1186/s13007-018-0275-x
https://pubmed.ncbi.nlm.nih.gov/29130192/

and thus such statements as "[...] this study demonstrates the feasibility and applicability [...]" should be clarified, if novel methodological aspects are actually presented, which I currently don't recognize, and the overall work should be better contextualized.

2. I would also strongly advise to make scripts, codes, and analyses related to this "workflow" freely available on platforms such as GitHub. In particular, it is not always clear what the authors have done, e.g. DGE done using all samples together, and then extracting contrasts, or first restricting to non-stromal, what are the covariates? etc.
In all cases, improving the methods description would certainly help, by providing clear descriptions of what was done for which analyses. Currently, all the results presented are hardly reproducible. The raw data generated 
in this study should also be made publicly available.

3. The way the data is presented is not entirely transparent. Unless I'm mistaken, the total number of patients used for RNA-seq is introduced first at line 186 (10 NSCLC, 6 LUAD, and 4 LUSC), only after the Methods section. Now, for ChIP-seq, only 3 LUAD, and 3 LUSC were used. Why? Six normal lung tissue H3K4me3 ChIP-seq samples were then collected from ENCODE, instead of using the normal tissue collected at the same time as the lung cancer tissue used for LMD isolation. Why? A lot of batches. How all this was handled in the analyses? See also detailed comments below.

4. In general, the authors keep repeating methodological descriptions in the different results sections. This is not only redundant, but makes the text of the results more cumbersome. Detailed methodology should be given in the Methods, and the Results section should contain the results only. Similarly, the Discussion section is much too lengthy, and partly redundant.

5. I suggest English language revision: minor typos, semantics. 

I have a few more detailed comments:

Fig 1b, abbreviations N, T, S are not introduced. Why this particular example (cherry picking)? A more global representation of the data would be more suitable. What is the use of associated Supplementary Figure S1? 
What does it show in particular? In what does it support the results?

Fig 1c PCA. What does it mean to have squamous and adenocarcinoma (type) for normal and stromal samples (condition)?

Line 189 and following. Supplementary-Table-S1 appears sufficient for details. A figure would help to visualize the distributions, and avoid throwing numbers which are not useful for the interpretation of the results.

Section 3.1 and following. Please provide the full DGE results (FC, p-values, etc.) as supplementary material. Currently, only "selected genes" are given.

Line 246: The TCGA RNA-seq raw count data was downloaded from batch 101 [...]
See above, such details should be put in the Methods section. There are no explanations why this subset in particular was selected? And what about other relevant subsets, if any, of the TCGA?

Fig 2 e,f There are no legend, axis titles? y values? In general, there is not much effort invested in the figures. What is the use of associated Supplementary Figure S2? 

Line 305:  By merging peak regions from all samples [...]
How was this done? Calling consensus peaks is not trivial, and can be done in many different ways.

Line 311: We identified 645 promoters [...]
Where is the full table of results?

Sections 3.5, 3.6, and 3.7.
The correlation between proximal H3K4me3 regions and DEGs, or the association enhancers-targeted genes is interesting, but I have reservations about the overall conclusions, because of the low number of samples, combined to the fact that the normal samples for ChIP-seq are not "matched" to the cancer samples, as alluded above. 
In addition, I am of the opinion that there is a lack of methodological details, as also alluded above. For example, we don't even know what correlation was used (Spearmann, Pearson, etc.), and its significance, how covariates were treated, etc.

Has any other alternative test or statistical approach been used to validate or corroborate these findings, e.g. limma camera (competitive test to assess whether the genes in a given set are highly ranked in terms of differential expression, given the ChiP-seq peaks)?  

The authors also examined the pathways enriched by the DEGs associated with tumor-altered H3K4me3 histone marks, and report results about some cancer-related pathways. Here again I have reservations. These DEGs are more or less all associated with cancer, as shown e.g. in Section 3.2. The question is whether any random subset of these DEGs is enriched in some cancer-related pathway or relevant structural biological processes (and I suspect there will be many), and how much more significant is the association for the subset of those associated with tumor-altered H3K4me3 histone marks. The same reasoning applies for GO enrichment using the targeted genes found in Section 3.6.

Fig 3: again we see some "cherry picked" examples, such as EPPK1 (Figure 3a). Figure 3b could be improved, drawing a more global schematic of the workflow (Sections 3.5, 3.6, 3.7).

Author Response

We would like to thank the reviewer for their thoughtful feedback that improved the quality of our manuscript. Here (in the attachment), we respond to each of their comments as thoroughly as possible.

Reviewer 3 Report

In this manuscript the authors deal with sample heterogeneity in lung cancer clinical samples. Laser microdissection is suggested as a method to separate cancer cells from stromal and normal tissue. The isolated cell populations are subjected to RNA-seq and Chip-seq analysis. The results presented correlated well with the current knowledge for non-small cell lung cancer malignancy.

The stated aim of this paper is achieved. The authors present an easy to follow workflow of integrative analysis which is well demonstrated and sufficiently proven. The depth of discussion and knowledge of relevant bibliography is applauded. In my opinion this manuscript is well suited for a reputed cancer biology journal as "Cancers" and contributes to the field of personalized cancer therapeutics. I only have minor suggestions to make.

  1. From the material and methods section of the paper to the end, I found a well written manuscript. Abstract and introduction needs few English language improvements. On lines 24 and 55 derive could be replaced by drive. On line 36 the word "such" could be omitted. On line 45 the word "the" could be omitted. On line 85 the word "well" could be omitted.
  2. In section 2.4 it would be helpful if the lysis buffer and dilution buffer ingredients are mentioned.
  3. In figure 1b legend it would be helpful to note that N stands for normal T for tumor and S for stroma. Same for figure 2a, 3a, 3e and 4e.

This manuscript presents a sound methological pipeline for the analysis of lung cancer specimens genetic and epigenetic changes.

Round 2

Reviewer 2 Report

I have no further comments.